# Endothelium-Independent Vasodilatory Effects of Isodillapiolglycol Isolated from *Ostericum citriodorum*

**DOI:** 10.3390/molecules25040885

**Published:** 2020-02-17

**Authors:** Tengshuo Luo, Zewei Chen, Fengyun Wang, Shanshan Yin, Pan Liu, Jun Zhang, Zhonghua Yang

**Affiliations:** 1School of Pharmaceutical Sciences, Guangzhou University of Chinese Medicine, Guangzhou 510006, China; luots512@163.com (T.L.); czwbs007@163.com (Z.C.); 2Guangdong Pharmaceutical University, Guangzhou 510006, China; wfycn2000@163.com; 3HEC Pharm R&D Center, Dongguan 523871, China; shansy6@163.com (S.Y.); gzliupan@163.com (P.L.); 4Medical College of Acupuncture Moxibustion and Rehabilitation, Guangzhou University of Chinese Medicine, Guangzhou 510006, China

**Keywords:** *Ostericum citriodorum*, isodillapiolglycol, vasodilation, endothelium-independent, Ca^2+^ channels

## Abstract

*Ostericum citriodorum* is a plant with a native range in China used in herbal medicine for treating angina pectoris. In this study, we investigated the vasodilatory effects of isodillapiolglycol (IDG), which is one of the main ingredients isolated from *O. citriodorum* ethyl acetate extract, in Sprague–Dawley rat aortic rings, and measured intracellular Ca^2+^ ([Ca^2+^]_in_) using a molecular fluo-3/AM probe. The results show that IDG dose-dependently relaxed endothelium-intact or -denuded aortic rings pre-contracted with noradrenaline (NE) or potassium chloride (KCl), and inhibited CaCl_2_-induced contraction in high K^+^ depolarized aortic rings. Tetraethyl ammonium chloride (a Ca^2+^-activated K^+^ channel blocker) or verapamil (an L-type Ca^2+^ channel blocker) significantly reduced the relaxation of IDG in aortic rings pre-contracted with NE. In vascular smooth muscle cells, IDG inhibited the increase in [Ca^2+^]_in_ stimulated by KCl in Krebs solution; likewise, IDG also attenuated the increase in [Ca^2+^]_in_ induced by NE or subsequent supplementation of CaCl_2_. These findings demonstrate that IDG relaxes aortic rings in an endothelium-independent manner by reducing [Ca^2+^]_in_, likely through inhibition of the receptor-gated Ca^2+^ channel and the voltage-dependent Ca^2+^ channel, and through opening of the Ca^2+^-activated K^+^ channel.

## 1. Introduction

Angina pectoris (AP) is defined as a clinical syndrome of temporary and rapid myocardial ischemia or anoxia caused by persistent coronary stenosis or spasm [1,2]. The mechanisms underlying AP might be multifactorial. NO released from vascular endothelial cells causes endothelium-dependent vasodilation, and significant endothelial damage might result in vasoconstriction and may induce AP [3,4]. Hypercontractility of vascular smooth muscle cells (VSMCs) has been shown to be strongly associated with AP, and Ca^2+^ is the primary regulator of tension in VSMCs. There are two main types of channel in VSMCs: the receptor-operated Ca^2+^ channel (ROCC) and the voltage-dependent Ca^2+^ channel (VDCC). Activation of ROCC causes extracellular Ca^2+^ influx and intracellular Ca^2+^ ([Ca^2+^]_in_) release from the sarcoplasmic reticulum (SR), and activation of VDCC causes extracellular Ca^2+^ influx [5,6]. Until now, NO donor drugs (nitroglycerin) or Ca^2+^ channel antagonists (nifedipine) have mainly been used to treat AP in clinical practice [7,8]. Although efficient for treating AP symptoms to a certain extent, these agents do not provide relief in all patients with different conditions; therefore, a wider range of therapeutic agents is needed to treat AP than is currently available [9].

*Ostericum citriodorum* is a herbal medicine used to treat AP, coronary disease, and hypertension in China [10,11,12]. Its effect have been clearly recorded in China’s Handbook of Common Chinese Herbal Medicine, National Compendium of Chinese Herbal Medicine, Cihai of Traditional Chinese Medicine, and Chinese Materia Medica dictionary. For example, the Huxin capsule, which is composed of *O. citriodorum*, has a good effect for treating AP in clinical practice [13].

In our previous study, we found that isoapiole, which was separated from the petroleum ether extract of *O. citriodorum*, relaxed the aortic rings in an endothelium-dependent manner [13]. A diversity of active ingredients and pathways are found in medicinal plants [14], and we further reported that the ethyl acetate extract of *O. citriodorum* exhibited an endothelium-independent vasodilatory effect [15,16]. In this research, we obtained isodillapiolglycol (IDG), one of the main ingredients from the *O. citriodorum* ethyl-acetate extract (OCE). On the basis of IDG possessing vasorelaxant activity, we investigated the mechanism of its vasodilatation by regulating [Ca^2+^]_in_ in VSMCs.

## 2. Results

### 2.1. Vasodilatory Effect of the Eight Major Fractions Obtained from O. Citriodorum Ethyl-Acetate Extract (OCE)

In this primary experiment, we examined the vasodilatory effects of eight major fractions (Fr. A–H) obtained from OCE. The aorta rings were pre-contracted with noradrenaline (NE) (3 μM), before adding Fr. A–H (1000 μg/mL). Among the eight fractions, Fr. F had the strongest vasodilatory effect (Figure 1).

### 2.2. Isolation and Identification

The molecular formula of the compound obtained from Fr. F was C_12_H_12_O_6_, electron ionization mass spectrometry (EI–MS) (positive) *m*/*z*: 279.085 [M + Na] ^+^. ^1^H-NMR (CDCl_3_, 400MHz): *δ*_H_ 6.48 (1H, s, H6), 5.96 (2H, s, H4′), 4.52 (1H, d, *J* = 7.2 Hz, H1′), 3.85 (1H, m, H2′), 3.92 (3H, s, 2—OCH_3_), 3.85 (3H, s, 3—OCH_3_), 1.07 (3H, d, *J* = 6.4 Hz, H3′). In the heteronuclear multiple bond correlation (HMBC)spectrum, we found correlations between H1′ and C1/C2/C6, and between H2′ and C1, indicating that C1 was connected to C1′. The HMBC correlations between H4′ and C4/C5 suggested the connection of C4, C5, and the methylene dioxy groups of C4′ via an oxygen bond, and the position of the methoxy group could be defined by the HMBC correlations between H2–OCH_3_ and C2, and between H3-OCH_3_ and C3. We found HMBC correlations between H6 and C1/C2/C4/C5. Thus, the planar structure was established (Figure 2A). Finally, we identified an erythro isomer between H1′ and H2′ caused by the coupling constant of H1′ and H2′, *J* = 7.2 Hz (Figure 2B). Based on the above information, we determined that this compound is 1′,2′-erythro-1′,2′-dioxy-2,3-dimethoxy-4,5-methylenedioxy-1-propylbenzene.

The ^1^H-NMR data were basically consistent with those reported previously [17]; thus, the compound was identified as ^13^C-NMR (CDCl_3_, 100 MHz) data: *δ*_C_ 139.1 (C3), 138.4 (C4), 136.6 (C5), 136.0 (C2), 126.1 (C1), 106.8 (C6), 101.9 (C4′), 75.6 (C1′), 71.8 (C2′), 60.1 (3—OCH_3_), 57.2 (2—OCH_3_), 18.9 (C3′).

By optimizing the chromatographic conditions, a Diamonsil C_18_ column (5 μm, 250 × 4.6 mm) was determined as the stationary phase. The mobile phase was 35% methanol–water detected at 210 nm. We detected that the purity of IDG was 97.0% and the content in *O. citriodorum* was 0.2955 mg·g^−1^ (retention time = 25.05 min; Figure 2C).

In addition to IDG being isolated from Fr. F, 8-(3,7-Dimethyl-octa-2,6-dienyl)-7-hydroxy-6-mehtoxy-chromen-2-one (1) was isolated from Fr. B. Decursidin (2) (CAS: 23027-48-7) and 9-angeloyloxy-10-senecioyloxy-9,10-dihydroxanthyletin (3) were isolated from Fr. C. Additionally, 6,7-Dimethoxy-1,3-benzodiox-ole-4-methanol (4) (CAS: 245421-64-1) was isolated from Fr. E. Nodakenetin (5) (CAS: 495-32-9) was isolated from Fr. F. Lariciresinol (6) (CAS: 27003-73-2) was isolated from Fr. G. Ostercitriodin A (7) was isolated from Fr. H (Figure 2D) (see Appendix A for compound details). Among them, 8-(3,7-Dimethyl-octa-2,6-dienyl)-7-hydroxy-6-mehtoxy-chromen-2-one and ostercitriodin A are the newly isolated compounds.

### 2.3. Vasorelaxant Effect of Isodillapiolglycol (IDG) vs. OCE at the Same Dose

We investigated the relaxation effect of OCE or IDG on the vasoconstriction induced by NE stimulation. The endothelium-denuded aortic rings were pre-contracted with NE (3 μM), and after this IDG (200, 400, 800 μg/mL) or OCE (200, 400, 800 μg/mL) was added. Within our designed dosage range, the vasorelaxant effect of IDG was significantly different from that of OCE (*p* < 0.01), which indicated the stronger vasodilation of IDG compared to that of OCE at the same dose (Figure 3).

### 2.4. IDG Inhibition of Contraction Induced by NE or KCl in Endothelium-Intact or -Denuded Aortic Rings

In these experiments, we investigated the vasorelaxant effects of IDG in aortic rings pre-contracted with NE (3 μM) or KCl (60 mM) in Krebs solution, and the half maximal effective concentration (EC_50_) was calculated.

At concentrations ranging from 0.7 to 2.8 mM, IDG was observed to remarkably relax the aortic rings pre-contracted with NE or KCl in a dose-dependent manner, and the EC_50_ values of the relaxing effect of IDG for endothelium-intact arteries and endothelium-denuded arteries were 1.48 ± 0.05 and 1.54 ± 0.1 mM in NE-induced contraction and 1.59 ± 0.04 and 1.52 ± 0.09 mM in KCl-induced contraction, respectively. The data suggested that the vasodilatation of IDG in the endothelium-intact aortic rings was not significantly different from that found in the endothelium-denuded rings (Figure 4). On the other hand, the IDG (0.7, 1.4, 2.8 mM) groups were statistically different compared to the sodium nitroprusside(SNP) group (^#^
*p* < 0.01).

### 2.5. IDG Relaxed High K^+^ Depolarized Aortic Rings Contracted by CaCl_2_

In the Ca^2+^-free and high K^+^ Krebs solution, the cumulative addition of CaCl_2_ (0.1–10 mM) induced a progressively increased vascular tension in aortic rings [18]. Pre-incubation with IDG (0.7, 1.4, or 2.8 mM) significantly inhibited the concentration-response contraction of CaCl_2_ (0.1–10 mM); compared with the control (CON) group, the maximum contraction was 64.48% ± 13.61%, 56.38% ± 5.19%, and 53.73% ± 2.67%, respectively (Figure 5).

### 2.6. Verapamil (VER) Inhibition of Vasorelaxant Effects of IDG in Aortic Rings Pre-Contracted with NE

The L-type Ca^2+^ channel (LTCC) is a major pathway for Ca^2+^ entry in most blood vessels, which can be inhibited by VER [19,20]. Pretreatment of endothelium-denuded aortic rings with VER significantly diminished NE-induced vasoconstriction (Figure 6A), after which IDG was added, eliciting vasodilatation. In the presence or absence of VER (10 μM), the maximal vasorelaxant effects were 24.51% ± 2.22% and 63.85% ± 6.09%, respectively. We suggest that IDG still had a vasodilating effect when the L-type Ca^2+^ channel was blocked; this vasorelaxant effect was significantly lower than that in the control group without VER, indicating that the vasodilatation of IDG was blocked by VER (Figure 6B).

### 2.7. Tetraethylammonium-Chloride Inhibition of Vasorelaxation of IDG in NE Pre-Contracted Aortic Rings 

In NE-induced vasoconstriction, pretreatment with tetraethylammonium chloride (TEA; 1 mM), a blocker of the Ca^2+^-activated K^+^ channel (K_Ca_), reduced the vasodilatory effects of IDG. The maximum contraction (E_max_) of IDG was 23.76% ± 3.07% (EC_50_ = 1.23 ± 0.15 mM) at a dose of 2.8 mM vs. the CON group’s value of 60.10% ± 4.57%. However, BaCl_2_, a blocker of the inward-rectifier potassium ion channel (K_ir_), did not affect the vasodilatory effects of IDG. In the presence and absence of BaCl_2_ (10 μM), the E_max_ values with IDG preincubation (2.8 mM) were 57.9% ± 3.19% (EC_50_ = 1.55 ± 0.04 mM) and 60.10% ± 4.57% (EC_50_ = 1.47 ± 0.01 mM), respectively (Figure 7).

### 2.8. IDG Reduced the Increase in [Ca^2+^]_in_ Stimulated by NE or KCl

We used a fluo-3/AM molecular probe to observe the change of [Ca^2+^]_in_ fluorescence intensity in VSMCs. In the first instance, the increase of [Ca^2+^]_in_ fluorescence intensity generated by NE in VSMCs pre-treated with IDG was significantly lower (*p* < 0.05) in Ca^2+^-free Krebs solution than that observed in the control (Figure 8). Then, the addition of CaCl_2_ (2.5 mM) in VSMCs of these groups showed a significant elevation in the [Ca^2+^]_in_ fluorescence intensity; however, compared with CON group, the [Ca^2+^]_in_ fluorescence intensity in VSMCs pre-treated with IDG (0.7, 1.4, and 2.8 mM) produced decreases of 11.93%, 24.23% and 65.95%, respectively (Figure 8B). In another group, a similar phenomenon was observed when VSMCs were stimulated by KCl (100 mM) in Krebs solution, with the [Ca^2+^]_in_ fluorescence intensities decreasing by 13.84%, 34.82%, 52.58%, and 43.18% compared with the CON group after the cells were preincubated with IDG (0.7, 1.4, and 2.8 mM) and SNP (0.1 μM), respectively (Figure 8).

## 3. Discussion

IDG is one of the main major constituents of *O. citriodorum*. This study demonstrated that IDG showed a vasorelaxant effect in isolated rat aortic rings, which occurred in a dose-dependent and endothelium-independent manner. More importantly, we determined the mechanism of IDG vasorelaxation through reducing [Ca^2+^]_in_ in VSMCs and found that it was probably related to ROCC, VDCC, and K_Ca_.

The OCE was subjected to a silica gel column using gradient mixtures of chloroform–methanol as eluents to obtain eight major fractions (Fr. A–Fr. H). We investigated the vasodilating effect of these fractions and found that Fr. F had the strongest vasodilatory effect. Then, we further separated and purified these fractions. Using Sephadex LH-20 columns or ostade-cylsilane (ODS)columns, we obtained IDG and 7 other compounds, among which 8-(3,7-Dimethyl-octa-2,6-dienyl)-7-hydroxy-6-mehtoxy-chromen-2-one and ostercitriodin A are the newly isolated compounds. In a preliminary experiment, IDG, which was obtained from Fr. F, showed a stronger vasodilatory effect than other compounds. We then investigated the vasorelaxant effects of IDG in rat aortic rings and the possible mechanisms of this action.

It is known that NE binds to the serpentine receptor and activates phospholipase C through G-protein, which generates inositol-1,4,5- triphosphate (IP3) [21]. The binding of IP3 to receptors on the SR results in the release of Ca^2+^ into the cytoplasm, and activates ROCC on the cell membrane, leading to extracellular Ca^2+^ influx [22,23,24]. A high concentration of KCl induced cell membrane depolarization and activated VDCC [25]. Our findings showed that IDG (0.7, 1.4, and 2.8 mM) dose-dependently relaxed NE- or KCl-induced contraction in endothelium-intact or -denuded aortic rings, and also inhibited CaCl_2_-induced contraction in high K^+^ depolarized aortic rings. The results show that IDG exerted vasodilating effects in a dose-dependent and endothelial-independent manner, and possibly also through inhibiting extracellular Ca^2+^ influx. On the other hand, the vasoconstriction stimulated by NE could still be relaxed by IDG, whether or not aortic rings were pre-incubated with VER. However, the group with VER was weaker than CON group, which means that (1) IDG may play a role in inhibiting the release of internal Ca^2+^ and that (2) the vasodilatory effect of IDG was related to the L-type Ca^2+^ channel.

Ca^2+^ is a critical factor in excitation–contraction coupling in VSMCs [22,23]. To prove that IDG relaxed the aortic rings by regulating [Ca^2+^]_in_ in VSMCs, a molecular fluo-3/AM probe was used to measure real-time changes in [Ca^2+^]_in_ induced by KCl or NE. In the Ca^2+^-free Krebs solution, the average fluorescence intensity of VSMCs was detected after stimulation by NE and subsequent supplementation of CaCl_2_. The average mean fluorescence intensities of different doses of IDG groups significantly decreased compared with the CON group, indicating that IDG inhibited the extracellular Ca^2+^ influx and intracellular Ca^2+^ release through ROCC. Different concentrations of IDG attenuated the increase in [Ca^2+^]_in_ fluorescence intensity induced by KCl, which suggests that IDG could act on VDCC, inhibiting Ca^2+^ influx. Therefore, these findings demonstrated that IDG inhibits the KCl- or NE-induced increases in [Ca^2+^]_in_ in VSMCs, possibly through VDCC and ROCC.

Besides Ca^2+^ channels, K^+^ channels also play an important role in vasodilation [26,27]. Activation of K^+^ channels in VSMCs normally hyperpolarizes the cell membrane due to an efflux of K^+^, which suppresses VSMC contraction [28,29]. Our experiment demonstrated that the effects of IDG in NE-contracted rings were attenuated by TEA but not BaCl_2_, which manifested in the vasodilatory effects related to K_Ca_.

Taken together, our findings indicated that IDG caused endothelium-independent vasorelaxation in rat aortic rings, and that the vasorelaxant activity was related to ROCC, VDCC, or K_Ca_ (Figure 9). Further studies are needed to investigate the precise mechanisms of its action.

A limitation of this study is the lack of research about IDG absorption in vivo. It is important to note that an ex vivo model of isolated aortic rings is a useful system for testing the vasoreactivity potency of compounds, but does not allow comprehensive evaluation of the potential pharmacological activities or functions [30]. In terms of the efficacy or the future therapeutic applications of IDG, systematically conducting in vivo or in vitro studies is necessary.

In our previous research [13], we found that isoapiole, one of the main ingredients isolated from the petroleum ether extract of *O. citriodorum*, relaxed the aortic rings in an endothelium-dependent manner via the NO pathway. However, the vasorelaxant effects of IDG worked in an endothelium-independent manner by reducing [Ca^2+^]_in_ in VSMCs. Thus, one of the more significant findings to emerge from this study is that the vasorelaxant effects of *O. citriodorum* are possibly derived from its variety of components, which exert their effects on disease by binding multiple targets, resulting in synergistic therapeutic activities.

Therapeutic agents sourced from herbal medicines show potential for the treatment of AP [31]. As a traditional herbal medicine, *O. citriodorum* has good clinical value for treatment of AP or coronary disease, but there are a few studies on this subject. To the best of our knowledge, this study is the first to analyze the OCE and its eight compounds, which included two newly isolated compounds. On the other hand, we have investigated the vasodilation effect of the 8 fractions (Figure 1). Both of the compounds showed medicinal effects, meaning that *O. citriodorum* has great potential for treating AP and other cardiovascular diseases. However, our experiment focuses on the research on IDG, which was isolated from Fr. F fractions, although this will help us to understand the role of *O. citriodorum* in treating AP. However, it is necessary to continue research on the vasodilation effect of the other components in *O. citriodorum* and explore the laws of their interaction. Not only could this explain the mechanism of *O. citriodorum* in treatment for AP and other diseases, but could also provide valuable leads for the development of natural drugs that are useful in the treatment of cardiovascular diseases.

## 4. Materials and Methods

### 4.1. Animals

Specific pathogen-free (SPF) grade healthy male Sprague–Dawley (SD) rats weighing 250 ± 30 g each were obtained from the Experimental Animal Center of Guangzhou University of Chinese Medicine (No: SCXK-2013-0020). The animals were caged under standard laboratory conditions, at a constant temperature (22 ± 1 °C) and under a 12 h light/dark cycle, with free access to food and water. The rats were also allowed to acclimatize to the animal facility for at least seven days before the start of the experiments. The animal care and use protocol was reviewed and approved by the Ethics Committee of Guangzhou University of Chinese Medicine (Date 6 March 2018, Approval number: 20180306022).

### 4.2. Drugs and Reagents

*O. citriodorum* was purchased from the Chinese medicine port of Guangxi Yulin, and was authenticated by Professor Haibo Huang at Guangzhou University of Chinese Medicine. Herbarium voucher specimens of *O. citriodorum* (XZ 08) were prepared and deposited at the Research and Development Center for new Chinese drugs, Guangzhou University of Chinese Medicine. NE, acetylcholine (Ach), DMSO, EGTA, TEA, and bovine serum albumin (BSA) were purchased from Sigma-Aldrich (St. Louis, MO, USA). Verapamil hydrochloride injection was purchased from Shanghai Harvest Pharmaceutical Co., Ltd. (Shanghai, China, 43141201). Sodium nitroprusside for injection was obtained from Hunan Koren Pharmaceutical Co., Ltd. (Hunan, China, B15070204). Dulbecco’s modified Eagle’s medium (DMEM) and fetal bovine serum (FBS) were obtained from Hyclone (Logan, UT, USA). Fluo-3-AM was acquired from Invitrogen (Carlsbad, CA, USA).

### 4.3. Cells

A7r5 cells were obtained from Shanghai Institute for Biological Sciences. The bath concentration of DMSO did not exceed 0.3%, which had no effect, per se, on the basal tonus of the preparations or on the agonist-mediated contraction or relaxation. The cell was cultured in Dulbecco’s modified Eagle’s medium containing 10% fetal bovine serum (FBS) at 37 °C with 5% CO_2_, and the medium was changed every 2 or 3 days.

### 4.4. Preparation of SD Rat Thoracic Aortic Rings 

Sprague–Dawley rats were narcotized using 10% chloral hydrate, and the abdominal aorta was bled to death. The thoracic aorta was dissociated and immediately transferred to ice-cold modified Krebs solution, then segmented into rings measuring approximately 3–4 mm each after clearing perivascular fat and adhesive connective tissues. Rings were mounted with two wire hooks: one of the hooks was fixed at the bottom of the organ bath, whereas the other was connected via a micrometric manipulator to a force displacement transducer for measurement of the isometric force. They were suspended in organ baths containing 15 mL Krebs solution filled with a continuous gas mixture of 95% O_2_ and 5% CO_2_ at 37 °C. The altered responses of the thoracic aorta tension were recorded with a ML870 Power Lab Biological Signal Collection System (AD Instruments, Castle Hill, NSW, Australia). The resting tension was gradually adjusted to 1.5 g over 1 h, and equilibrated for 30 min. The Krebs solution was replaced every 20 min.

### 4.5. Vasorelaxant Effects of Major Fractions in Aortic Rings Contracted by NE

In the first experiment, NE (3 μM) was used to induce a steady contraction in denuded endothelium aortic rings, and 8 major fractions (Fr. A–H) were added individually. The changes in vascular tension were recorded, and the vasodilation rate (%) was calculated as: Relaxation (%) = (maximal contraction by NE − tension at the corresponding time after incubation with tested compounds)/(maximal contraction by NE − basal tension) × 100%.(1)

### 4.6. Extraction of Compounds and Structural Analysis

Air-dried and powdered *O. citriodorum* (9 kg) was extracted with 80% ethanol (45 L) three times under heating reflux. The ethanol extract was concentrated under a vacuum to yield a crude extract, which was suspended in water and then extracted successively with petroleum ether and ethyl acetate. The ethyl acetate extract fraction (153.7 g) was subjected to a silica gel column using gradient mixtures of chloroform–methanol as eluents to obtain the 8 major fractions (Fr. A–H). Then, in Sephadex LH-20 columns (chloroform–methanol, 1:1, *v*/*v*), Fr. B (12.07g) was further subjected to an ODS column using MeOH−H_2_O (7:3, v/v) as an eluent to produce compound **1** (13 mg). Fr. C (3.39 g) was further subjected to an ODS column using MeOH−H_2_O (75:25, *v*/*v*) as an eluent to produce compounds **2** (5.2 mg) and **3** (3.7 mg). Fr. E (1.01 g) was separated using a Sephadex LH-20 column (CHCl_3_−MeOH, 50:50, *v*/*v*) to obtain three subfractions (Fr. E1–Fr. E3). Fr. E3 was further subjected to an ODS column using MeOH−H2O (7:3, *v*/*v*) as an eluent to produce compound **4** (7.8 mg). Fr. F (8.27 g) was separated using a Sephadex LH-20 column (CHCl_3_−MeOH, 50:50, *v*/*v*) to obtain seven subfractions (Fr. F1–F7). Fr. F6 was further purified by reversed-phase preparative HPLC using MeOH–H_2_O (3:7) as the mobile phase to yield **5** (22 mg) and **6** (7.5 mg). Fr. G (7.53 g) was separated using a Sephadex LH-20 column (CHCl_3_−MeOH, 50:50, *v*/*v*) to obtain five subfractions (Fr. G1–G7). Fr. G3 was separated by a Sephadex LH-20 column (MeOH−CHCl_3_, 5:95 to 95:5, *v*/*v*) to produce Fr. G3a−G3c. Fr. G3c was further subjected to an ODS column using MeOH−H_2_O (3:7, *v*/*v*) as an eluent to produce compound **7** (15 mg). Fr. H (5.16 g) was separated using a Sephadex LH-20 column (CHCl_3_−MeOH, 50:50, *v*/*v*) to produce five subfractions (Fr. H1–H5). Fr. H3 was separated using a Sephadex LH-20 column (MeOH−CHCl_3_, 5:95 to 95:5, *v*/*v*) to produce Fr. H3a−H3c. Fr. H3c was further subjected to an ODS column using MeOH−H_2_O (1:1, *v*/*v*) as an eluent to produce compound **8** (8.2 mg). The structures were established using a combination of spectroscopic methods, and the data were compared with those reported in the literature.

### 4.7. Vasorelaxant Effects of IDG in Aortic Rings Contracted by NE or KCl

To investigate the vasorelaxant activity, NE (3 μM) or KCl (60 mM) was used to induce a steady contraction in aortic rings with an intact or denuded endothelium, and IDG at three different doses (0.7, 1.4, or 2.8 mM) or SNP (100 nM) was added for 10 min to confirm inhibitory activities. The changes in vascular tension were recorded, and the vasodilation rate (%) was calculated as:Relaxation (%) = (the maximal contraction by NE (or KCl) − tension at the corresponding time after incubation with the tested compounds)/(maximal contraction by NE (or KCl) − basal tension) × 100%.(2)

To investigate the role of extracellular Ca^2+^ influx, LTCC, in the K^+^ channel in IDG-induced vasorelaxation, the following drugs or inhibitors were used: CaCl_2_, VER (10 mM), a blocker of LTCC; TEA (1 mM), a blocker of K_Ca_; BaCl_2_ (10 µM), a blocker of K_ir_. The endothelium-denuded aortic rings were pre-incubated with each blocker before contraction by NE (3 µM).

### 4.8. Effects of IDG on [Ca^2+^]_in_ in VSMCs

VSMCs were plated on dishes and loaded with Fluo-3/AM 10 µmol/L for 45 min in Krebs or Ca^2+^-free Krebs solution. VSMCs were then washed thrice with Krebs solution at 37 °C to remove the extracellular excess of dye. Excitation and emission wavelengths were set to 488 and 525 nm, respectively. The effects of IDG (0.7, 1.4, 2.8 mM) on KCl (100 mM)-induced fluorescence emission were evaluated in Krebs solution, and the effects of IDG on NE (10 µM)- and CaCl_2_ (2.5 mM)-induced fluorescence emission were evaluated in Ca^2+^-free Krebs solution. SNP (100 nM) was used as a positive control group and DMSO (0.27%) solution was used as a CON group. Changes in fluorescence were calculated as *F − F’*, where *F* is the maximum fluorescence intensity of the CON group and *F’* is the maximum fluorescence intensity of the drug group.

### 4.9. Statistical Analysis

Results were analyzed with GraphPad Prism 5 version 5.01 for Windows, (GraphPad Software Inc., La Jolla, CA, USA) using two-way ANOVA followed by Bonferroni’s post-hoc test. Curve fitting in the figures was generated using the same software using non-linear regression, and EC_50_ and E_max_ were compared using unpaired Student’s *t*-test. All the data are expressed as means ± SEM. Statistical significance was measured as *p* < 0.05 compared with the CON group.

## 5. Conclusions

In conclusion, our major novel finding is that IDG, which is one of the main ingredients isolated from OCE, relaxed aortic rings in an endothelium-independent manner, and we provided evidence that the mechanism of its vasorelaxation was via reducing [Ca^2+^]_in_ in VSMCs. Therefore, IDG shows promise for the development of subsequent Ca^2+^ channel blockers. However, further studies are needed to investigate the precise mechanisms of IDG action and its pharmacokinetics in vivo.

At the same time, we only studied the vasodilation effect of IDG, and the research on other compounds in *O. citriodorum* need further development. Previous preliminary experiments have shown that these components also have a certain vasodilation effect, meaning *O. citriodorum* or other medicinal ingredients have great potential for treating angina pectoris or other cardiovascular diseases, which deserves intensive scientific exploration.

## Figures and Tables

**Figure 1 molecules-25-00885-f001:**
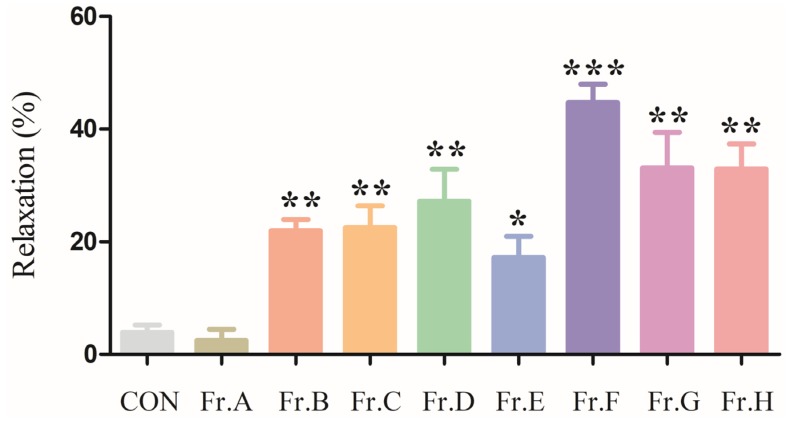
The vasorelaxant effect of fraction F (Fr. F) was stronger than those of the other major fractions obtained from the *Ostericum citriodorum* ethyl acetate extract (OCE). Endothelium-denuded aorta rings were pre-contracted with noradrenaline (NE) before the eight major fractions (Fr. A–H) (1000 μg/mL) were added. Values are means ± standard error of the mean (SEM) (*n* = 6). Note: * *p* < 0.05, ** *p* < 0.01 each fraction compared with control (CON, incubation the vascular rings with 0.2% DMSO) group using two-way ANOVA.

**Figure 2 molecules-25-00885-f002:**
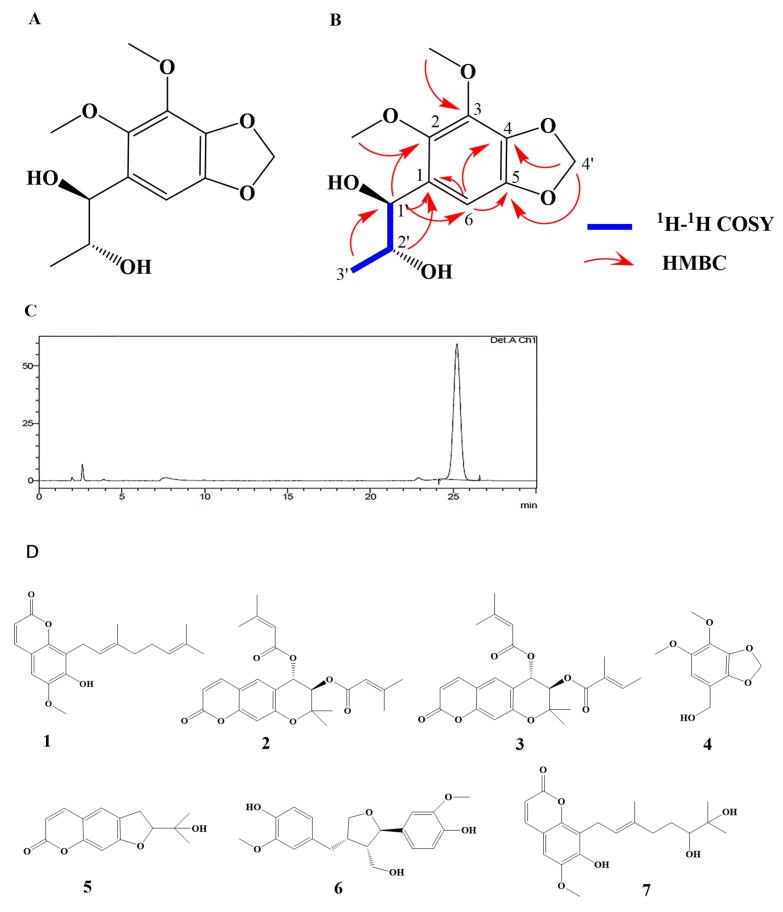
(**A**) Chemical structure, (**B**) key homonuclear chemical shift correlation spectroscopy (^1^H−^1^H COSY), and heteronuclear multiple bond correlation (HMBC)correlations. (**C**) HPLC chromatogram analysis of isodillapiolglycol (IDG). (**D**) Chemical structures of compounds **1**–**7**.

**Figure 3 molecules-25-00885-f003:**
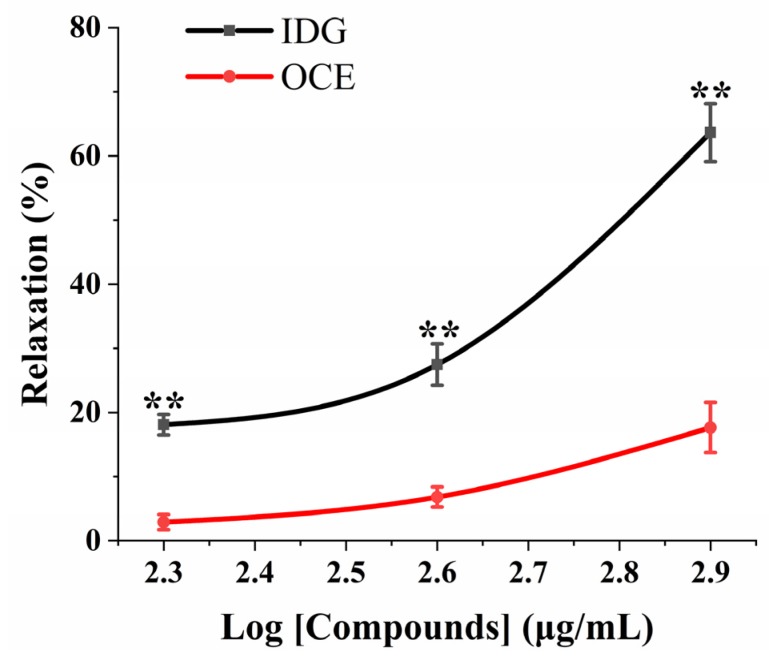
The vasodilatation induced by isodillapiolglycol (IDG) was stronger than that of OCE at the same dose. Endothelium-denuded aorta rings were pre-contracted with NE before the addition of IDG (200, 400, 800 μg/mL) or OCE (200, 400, 800 μg/mL). Values are means ± SEM (*n* = 6). Note: ** *p* < 0.01 compared with the OCE group using two-way ANOVA.

**Figure 4 molecules-25-00885-f004:**
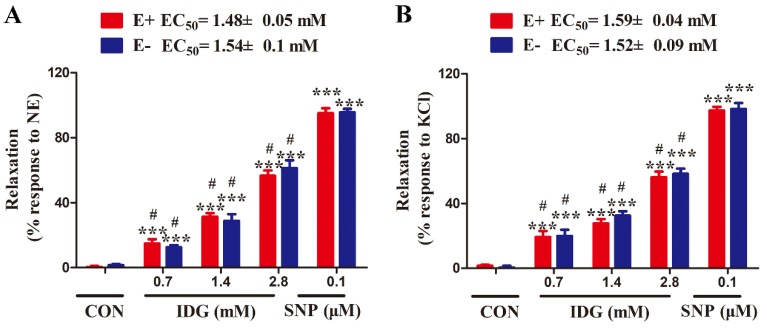
IDG directly caused relaxation in the vascular smooth muscle cells of aorta rings. Endothelium-intact (E+) or -denuded (E−) aorta rings were incubated with IDG (0.7, 1.4, or 2.8 mM) after being pre-contracted with (**A**) NE (3 μM) or (**B**) potassium chloride (KCl) (60 mM). Values are means ± SEM (*n* = 6). Relaxation for IDG was compared with control using two-way ANOVA. Note: *** *p* < 0.001 vs. CON (0.2% DMSO) group; ^#^
*p* < 0.01 vs. sodium nitroprusside (SNP) group.

**Figure 5 molecules-25-00885-f005:**
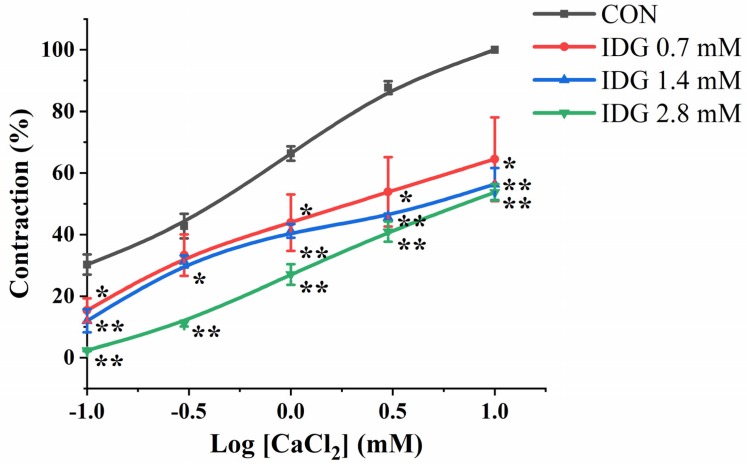
CaCl_2_-induced contractions of endothelium-denuded aortic rings pre-incubated in high K^+^ and Ca^2+^-free Krebs solution under conditions of pre-incubation with DMSO (negative control) or IDG (0.7, 1.4, 2.8 mM). Y-axis, % contraction compared to the contraction achieved with the highest Ca^2+^ concentration (10 mM) during the initial run without IDG in endothelium-denuded aortic rings. Values are mean ± SEM (*n* = 6). The contraction was determined using non-linear regression and a repeated-measures two-way ANOVA to compare curves. Statistically significant difference are * *p* < 0.05, ** *p* < 0.01, *** *p* < 0.001 vs. CON group.

**Figure 6 molecules-25-00885-f006:**
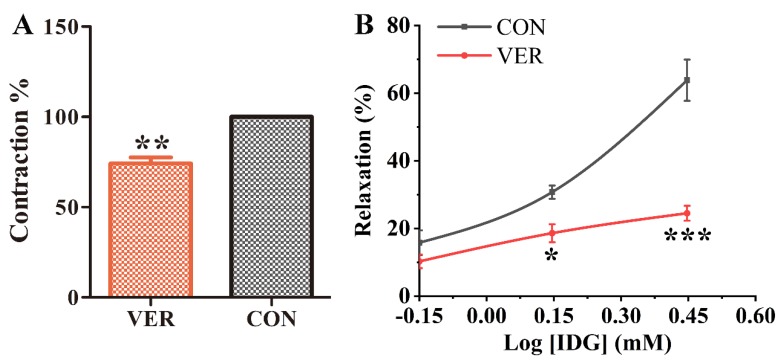
Effect of verapamil (VER) on the vascular relaxation of IDG. (**A**) In endothelium-denuded aortic rings contracted with NE before pre-incubation with or without VER (10 μM), the maximum contraction of the CON group caused by NE (10 mM) was recorded as 100%. (**B**) Vasoconstriction occurred when IDG (0.7, 1.4, or 2.8 mM) was added. Statistical analysis was performed using two-way ANOVA; * *p* < 0.05, *** *p* < 0.001 vs. CON group.

**Figure 7 molecules-25-00885-f007:**
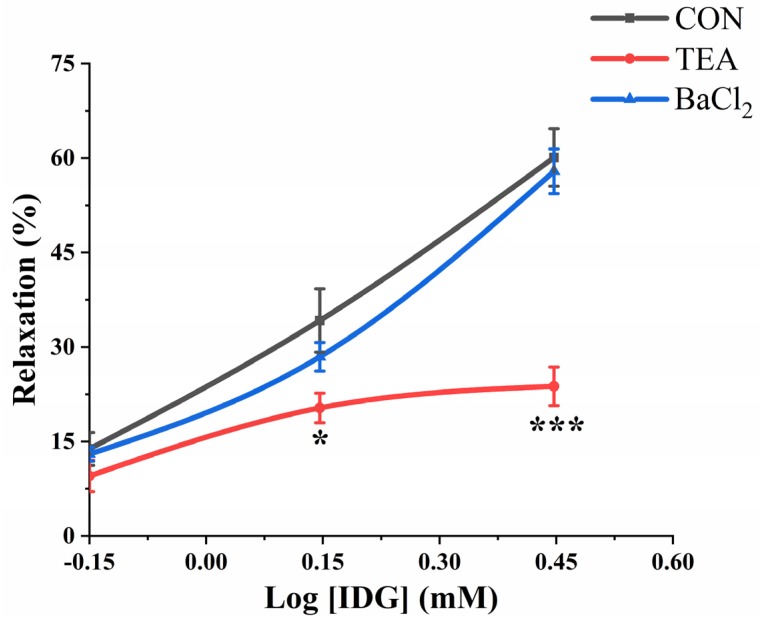
Evaluation of the IDG vasodilatation mechanism associated with K^+^ channels. Vasorelaxant effects of IDG (0.7, 1.4, or 2.8 mM) in NE-pre-contracted endothelium-denuded aorta rings with or without tetraethylammonium (TEA, 1 mM), a blocker of the Ca^2+^-activated K^+^ channel, or BaCl_2_ (10 μM), a blocker of the inward-rectifier K^+^ channel. The half maximal effective concentration (EC_50_)was determined using non-linear regression and a repeated-measures two-way ANOVA to compare curves; *n* = 6. Note: * *p* < 0.05, *** *p* < 0.001 vs. CON.

**Figure 8 molecules-25-00885-f008:**
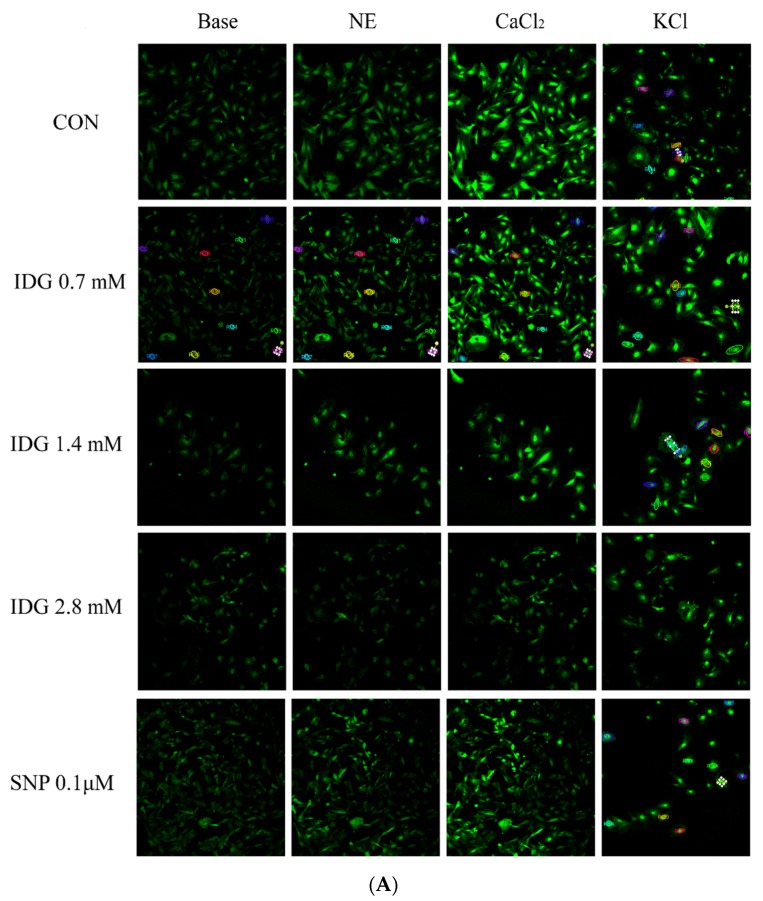
Relaxant effect of IDG (0.7, 1.4, and 2.8 mM) on vascular smooth muscle cells (VSMCs) contracted with NE (10 μM) or KCl (100 mM). (**A**) The change in fluorescence intensities represent the change in [Ca^2+^]_in_ levels. Original fluorescence images (40×). (**B**) The effects of IDG (0.7, 1.4, and 2.8 mM) reduced the increase in intracellular fluorescence intensity in VSMCs due to KCl (100 mM) in Krebs solution, NE (10 μM) in Ca^2+^-free Krebs solution, or the addition of CaCl_2_ (2.5 mM) to provide Ca^2+^. Values are means ± SEM (*n* = 6). Statistical analysis was performed using two-way ANOVA. Note: * *p* < 0.05, *** *p* < 0.001 compared with CON group.

**Figure 9 molecules-25-00885-f009:**
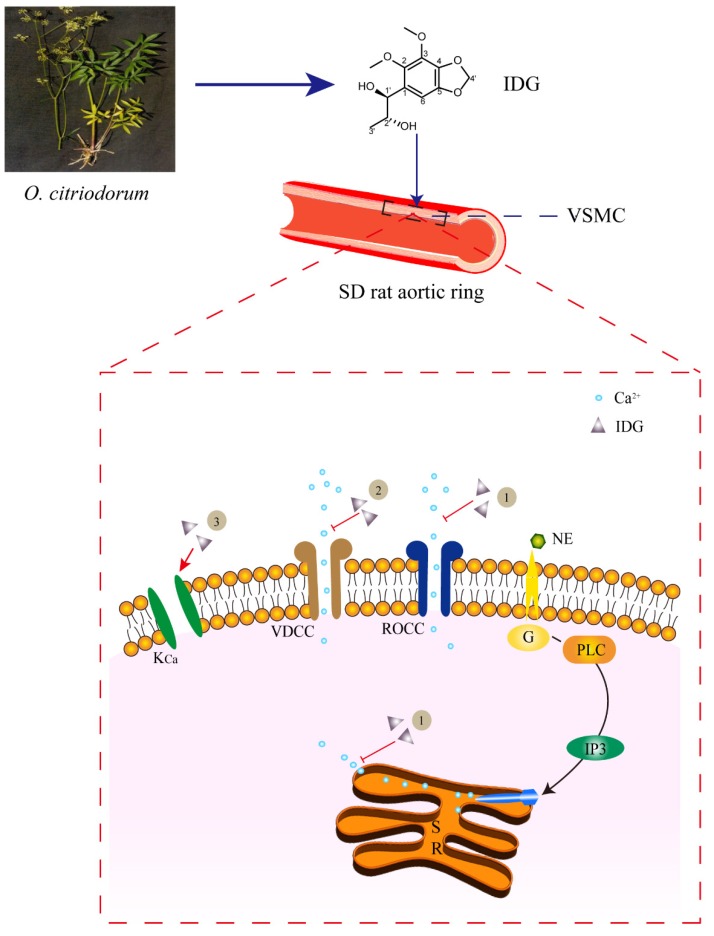
The role of IDG in relaxing vascular smooth muscle cells could be related to the following pathways: (**1**) the vasorelaxant effects of IDG caused by inhibiting extracellular Ca^2+^ influx and intracellular Ca^2+^ release through the receptor-operated Ca^2+^ channel (ROCC); (**2**) IDG-inhibited extracellular Ca^2+^ influx through the voltage-dependent Ca^2+^ channel (VDCC); (**3**) IDG may activate the K_Ca_ channel.

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
