# Peer review of "Endothelium-Independent Vasodilatory Effects of Isodillapiolglycol Isolated from Ostericum citriodorum"

_molecules, 2020, doi:10.3390/molecules25040885_

Round 1

Reviewer 1 Report

Overall, the authors responded to my questions. However, I wish to highlight the following issues:

If IDG is not absorbed from the intestine, it would not exert a function in in vivo Thus, I had suggested that the absorption of IDG should be described. This consideration can emphasize the possibility that IDG is a true candidate in O. citriodorum. The authors should discuss/add any information about this important point in the Discussion section. Alternatively, it is fair to say that the authors need to mention the necessity of investigating this issue in future studies. I had pointed out the authors’ incorrect understanding about the function of Ca2+-dependent K+ The authors still described in Fig. 9 that activation of Ca2+-dependent K+channels (KCa) induces Ca2+influx into the cells. KCacauses K+efflux out of the cells, resulting in hyperpolarization. Please check through the text again.

Author Response

Dear Reviewer,

Thank you very much for your advice. We have revised the paper and would like to re-submit it for your consideration.We answered your question in a Word and hope that the revision is acceptable.

Best Regards.

Yours sincerely,

Jun Zhang

Response to Reviewer 1 Comments

Point 1.If IDG is not absorbed from the intestine, it would not exert a function in in vivo Thus, I had suggested that the absorption of IDG should be described. This consideration can emphasize the possibility that IDG is a true candidate in O. citriodorum. The authors should discuss/add any information about this important point in the Discussion section. Alternatively, it is fair to say that the authors need to mention the necessity of investigating this issue in future studies.

Response: Our research team has conducted long-term and continuous research on O. citriodorum, we have isolated IDG, isoapiole and other compounds. For isoapiole, we have not only reported its vasodilation effect and mechanism [1], but also studied its pharmacokinetics[2]:UPLC-MS/MS technology was used to determine the pharmacokinetics of isoapiole in rat plasma, the results showed that: the half-life of isoapiole is 293.04 ± 29.59 min, Tmax is 360.00 ± 0.00 min; AUC0-t = 269.84 ± 77.70 μg / mL · min, AUC0 → ∞ = 282.71 ± 81.01 μg / mL · min, MRT0 -t = 550.80 ± 45.50min, and mainly absorbed through the intestine.

IDG and isoapiole are both phenylpropanoids, and IDG is a fat-soluble low-polar compound with a small molecular weight. Based on its chemical properties, we speculate that IDG has fast absorption, multi-tissue distribution, long residence time and other characteristics. We will conduct a systematic study of IDG's in vivo absorption process in the future. However, due to the large number of samples required to carry out the pharmacokinetic research of IDG, our research team needs to further purify the samples, so data cannot be obtained in a short time. We will speed up the progress of the experiment and report on it as soon as possible.

[1] Shanshan Yina, Shuangwei Zhang, Guoyong Tong, Lihong Deng, Tuliang Liang, Jun Zhang. In vitro vasorelaxation mechanisms of Isoapiole extracted from Lemonfragrant Angelica Root on rat thoracic aorta[J]. Journal of Ethnopharmacology, 2016, 188:229-233.

[2] Tuliang Liang, Rong Zhang, Chao Wang, Diling Liu, Muxia Li, Jun Zhang. UPLC-MS/MS analysis of active ingredient Isoapiole in rat plasma: a pharmacokinetic study [J]. PAK J PHARM SCI. (accepted).

Point 2. I had pointed out the authors’ incorrect understanding about the function of Ca2+-dependent K+ The authors still described in Fig. 9 that activation of Ca2+-dependent K+ channels (KCa) induces Ca2+ influx into the cells. KCa causes K+ efflux out of the cells, resulting in hyperpolarization. Please check through the text again.

Response: I'm sorry that we didn't correct this problem in time. We checked the relevant literature and studied again, and got the following conclusions [1][2]: Membrane hyperpolarization due to an efflux of K+ results from the opening of K+ channels in vascular smooth muscle. This effect is followed by the closure of voltage-dependent Ca2+ channels, leading to a reduction in Ca2+ entry, and vasodilation. However, our current experiments can only show that the relaxation of IDG was related to KCa channel, and how to further play the role needs to be studied. We have modified this part of the article and figure 9 based on your suggestions.

[1] Nelson M T , Quayle J M . Physiological roles and properties of potassium channels in arterial smooth muscle[J]. American Journal of Physiology, 1995, 268(1):799-822.

[2] Ko E A , Han J , Jung I D , et al. Physiological roles of K+ channels in vascular smooth muscle cells[J]. Journal of Smooth Muscle Research, 2008, 44(2):65-81.

Reviewer 2 Report

ABSTRACT

Abstract content changes are mainly related with English correction. Some indication on the IDG extraction must be in the abstract.

INTRODUCTION

English was improved. Despite the results section concerning chemical structure of IDG, the authors did not still indicate the relationship between isoapiole and IDG. I do not understand the relevance of mentioning this isoapiole compound, that is also vasorelaxant.

Some information concerning vascular contractility should be mentioned in the introduction, because this is an issue discussed to achieve the conclusions. Some introduction on extraction procedures will be also useful in this section.

RESULTS

- Figure 1 and pgs 62-70: The authors did not indicate what is the control group is (media??). This is a relevant information once the authors performed statistics based in the comparison with this group. In these experiments and other, further mentioned, the authors indicate that the rat aorta contraction was elicited by noradrenaline 3mM. This is a very big concentration, most of investigators use 1-10 µM as supramaximal concentration. Is this a mistake concerning units? (3 µM?)

- Lines 105-115; effect of IDG is bigger than that of OCE. Is this a not expected result? I think the authors must explain the meaning of this result in this section. A comparison with the effect of Fr. F will be useful in this section (but not strictly necessary).

- In general several figures are still pixeled in the pdf format.

- Figure 4: same question than Figure 1, The authors did not indicate what is the control group is.

- Figure 5. Most of authors show this type of graphics in the opposite sense, starting from -0.1 at the left of “x” axis (Log of CaCl2 concentration) and with 1.0 at the right of “x” axis. Like this contraction increases from left to right.  

- In my opinion, data of figure 6 do not demonstrate that IDG relaxes by inhibiting L type calcium channels; there are other hypothesis because both verapamil and IDG are vasodilators.  Anyway, it will be useful if the authors show a record of these experiments as figure.

The authors did not comment the SNP effects in the paragraph (lines 117-126).

DISCUSSION

Line 2013: NE coupled to G protein receptors…Noradrenaline is not coupled to G protein receptors.

A great part of the discussion is the explanation of basic vascular contractile events; maybe a great part of this information could be in the introduction. On the other hand, some discussion concerning the presence of IDG in O. citriodorum and the scientific demonstration about the potential use of this natural product is missing. Also the information/discussion concerning the possibility of extraction of IDG and others from O. citriodorum in order to obtain purified drugs is missing.

Lines 219-24: In this paragraph the author conclude that IDG acts by several mechanism, inhibiting LTCC and ROCC and by inhibiting the release of internal Ca2+, but the experiment performed did not demonstrate this because procedures to analyze inhibition of release of internal Ca2+ and to analyze ROCC were not performed in the study.

Figure 7 shows that IDG acts in VDCC, ROCC, KCa channels and also on exit of Ca2+ from endoplasmic reticulum. In my opinion, these are only possibilities or hypothesis, because the results shown does not demonstrate that these all are targets of IDG.

MATHERIAL & METHODS

- The information concerning A7r5 cells culture is mentioned in section 4.12 and not in 4.3 (media, passage, timing…). In my opinion this is not convenient.

- In this sense, this section has 13 subsections. In my opinion, there are only 3 types of experiments (extraction, contractility and intracellular calcium), 2 sample types (aorta rings and A7r5 cells), 1 set of chemicals (drugs and reagents) and statistics. Takin this in account, my opinion is that 7-8 subsections will be sufficient. The authors have 6 subsections starting by “Vasorelaxant Effects of..” or “Effects of…” that could be unified in 2 or maximum 3 subsections.

CONCLUSSIONS

- I agree with the vasodilation and “endothelium-independent” effect of IDG, and also with intracellular reduction in calcium (this was demonstrated by the data). But the authors conclude  “…involves two mechanisms: (1) inhibition of the entry of extracellular Ca2+ by blocking ROCC and VDCC and activating KCa, and (2) reducing [Ca2+]in release via inhibiting ROCC.” I think this are hypothesis to be confirmed because they are not completely demonstrated by the data.

- Some conclusions concerning the presence of IDG in O. citriodorum and the scientific demonstration about the potential use of this natural product are missing. Also the potential of the plant to extract this and other “new drugs”.

Author Response

Dear Reviewer,

Thank you very much for your advice. We have revised the paper and would like to re-submit it for your consideration.We answered your question in a Word and hope that the revision is acceptable.

Best Regards.

Yours sincerely,

Jun Zhang

Response to Reviewer 2 Comments

Point 1. Abstract content changes are mainly related with English correction. Some indication on the IDG extraction must be in the abstract.

Response: We have modified the content of the abstract based on your comments.

Point 2. English was improved. Despite the results section concerning chemical structure of IDG, the authors did not still indicate the relationship between isoapiole and IDG. I do not understand the relevance of mentioning this isoapiole compound, that is also vasorelaxant.

Response: Isoapiole is an effective component isolated from the petroleum ether part of O. citriodorum and has an endothelial-dependent vasodilation, IDG is an active ingredient obtained from the ethyl acetate extract of O. citriodorum and has an endothelium-independent vasodilation, both of them are phenylpropanoids. In our opinion, this is a very interesting discovery that two compounds (isoapiole and IDG) isolated from O. citriodorum have different vasodilating effects in different ways. Therefore, the isoapiole and IDG are enumerated here to reflect the multi-component and multi-target effect of O. citriodorum.

Point 3. Some information concerning vascular contractility should be mentioned in the introduction, because this is an issue discussed to achieve the conclusions. Some introduction on extraction procedures will be also useful in this section.

Response: We have made corresponding changes to the content of the article based on your comments.

Point 4. - Lines 105-115; effect of IDG is bigger than that of OCE. Is this a not expected result? I think the authors must explain the meaning of this result in this section. A comparison with the effect of Fr. F will be useful in this section (but not strictly necessary).

Response: As you said, it’s our expected results that the vasodilation effect of IDG is stronger than the OCE. Before we know whether the IDG has a vasodilating effect, we need to verify the vasodilation effect of IDG or it is one of the main active ingredients in OCE by experiments. The purpose of our entire study is to obtain the active components of vasodilation in OCE and to explore the mechanism of vasodilation. Therefore, after obtaining IDG from OCE, in order to examine the efficacy of IDG and IDG is the main component of OCE, we compared the vasodilation effect of IDG and OCE.

Point 5.- Figure 1 and pgs 62-70: The authors did not indicate what is the control group is (media??). This is a relevant information once the authors performed statistics based in the comparison with this group. In these experiments and other, further mentioned, the authors indicate that the rat aorta contraction was elicited by noradrenaline 3mM. This is a very big concentration, most of investigators use 1-10 µM as supramaximal concentration. Is this a mistake concerning units? (3 µM?)

Response: The control group indicate that the vascular rings were incubated with 0.2% DMSO. And is our carelessly written the wrong dose of NE, which should be 3 µM instead of 3 mM. At the same time, we have rechecked the data in the article, thank you for your correction.

Point 6. - Figure 4: same question than Figure 1, The authors did not indicate what is the control group is.

Response: We have modified the content of the abstract based on your comments, the control group indicate that the vascular rings were incubated with 0.2% DMSO.

Point 7. - Figure 5. Most of authors show this type of graphics in the opposite sense, starting from -0.1 at the left of “x” axis (Log of CaCl2 concentration) and with 1.0 at the right of “x” axis. Like this contraction increases from left to right.

Response: We have modified the figure based on your comments.

Point 8. - In my opinion, data of figure 6 do not demonstrate that IDG relaxes by inhibiting L type calcium channels; there are other hypothesis because both verapamil and IDG are vasodilators. Anyway, it will be useful if the authors show a record of these experiments as figure.

Response: I'm sorry that we haven't clearly expressed the content in this part. We all known that L-type Ca2+ channel is the main channel for extracellular Ca2+ inflow and VER is an L-type Ca2+ channel blocker. So, the experiment was divided into two groups. One group of vascular rings was pre-incubated with VER before NE added (VER group). The other group of vascular rings was directly stimulated with NE (control group). Then two groups of vascular rings were incubated with IDG and we recorded vascular ring tension after NE or IDG were added, respectively. 

If the degree of vasoconstriction stimulated directly with NE is recorded as 100%, the degree of vasoconstriction of NE stimulated vascular ring after pre-incubation with VER is only 74% (Figure 1A). After IDG incubation, it can be seen that the maximum degree of vasodilation in the control group can reach 63.85% ± 6.09%, while the degree of vasodilation in the VER group was 24.51% ± 2.22% (Figure 1 B).

We can get two experimental results:

(1) the vasoconstriction stimulated by NE can still be relaxed by IDG, whether or not aortic rings was pre-incubated with VER. NE-stimulated vasoconstriction is achieved by external Ca2+ influx and internal Ca2+ release, when the main channel of external Ca2+ inflow was blocked, the vasodilating effect of IDG may be related to inhibition of internal Ca2+ release.

(2) The vasodilating effect of IDG on vascular rings which was pre-incubated with VER was weaker than the control. When the channel of external Ca2+ inflow was blocked, the vasodilatory effect of IDG was reduced, too. So, we have reason to conclude that the vasodilatory effect of IDG is related to L-type Ca2+ channel. At the same time, we express these two results in the form of pictures.

In general, we have shown the relevance of this effect through isolated vascular ring experiments, further studies are needed to investigate the precise mechanisms of its action.

Figure 1. Effect of verapamil (VER) on the vascular relaxation of IDG. (A) In endothelium-denuded aortic rings contracted with NE before pre-incubation with or without VER (10 μM), the maximum contraction of the CON group caused by NE (10 mM) was recorded as 100%. (B) Vasoconstriction occurred when IDG (0.7, 1.4, or 2.8 mM) was added. Statistical analysis was performed using two-way ANOVA; *p < 0.05, ***p < 0.001 vs. CON group.

Figure 2. We want to express these through this picture: when both groups of vascular rings were stimulated by NE, the vasoconstriction of the control group is greater than the VER group; after incubated with IDG, both of vascular rings were relaxed, but the degree of the control group was stronger than the VER group.

Point 9. The authors did not comment the SNP effects in the paragraph (lines 117-126).

Response: Here we have labeled the analysis results of SNP and IDG in the figure, which are not listed in the text, but now have been changed.

Point 10. Line 2013: NE coupled to G protein receptors…Noradrenaline is not coupled to G protein receptors.

Response: Thank you for your correction of our question, we misrepresented this concept and we have changed it: NE binding to serpentine receptor (G Protein-Coupled Receptors) and activates phospholipase C through G-protein which generates inositol-1,4,5- triphosphate (IP3).

Point 11. A great part of the discussion is the explanation of basic vascular contractile events; maybe a great part of this information could be in the introduction. On the other hand, some discussion concerning the presence of IDG in O. citriodorum and the scientific demonstration about the potential use of this natural product is missing. Also the information/discussion concerning the possibility of extraction of IDG and others from O. citriodorum in order to obtain purified drugs is missing.

Response: We have modified the discussion of the article based on your comments.

Point 12. In this paragraph the author conclude that IDG acts by several mechanism, inhibiting LTCC and ROCC and by inhibiting the release of internal Ca2+, but the experiment performed did not demonstrate this because procedures to analyze inhibition of release of internal Ca2+ and to analyze ROCC were not performed in the study.

Response: NE activates receptor-operated Ca2+ channels (ROCC) to promote extracellular Ca2+ influx and release of internal Ca2+, thereby causing vasoconstriction.

In our experiments, IDG can relax vasoconstriction caused by NE stimulation, which means that the vasodilation effect of IDG may be related to ROCC.

In the cell experiments, we cultured the cells in Ca2+-free Kreb's solution, the administration group was given IDG, and the control group was given DMSO. When NE was added to the cells, the fluorescence intensity of the control group was significantly higher than the administration group, indicating that the presence of IDG inhibits the release of intracellular Ca2+ (because NE will only promote intracellular Ca2+ release through the ROCC to increase intracellular Ca2+ concentration). Then, we provide the cells with an extra Ca2+ environment by adding CaCl2. We can observe that the fluorescence intensity of the cells in each group increased significantly, but the expression of the fluorescence intensity of the cells in the control group was significantly higher than the administration group, indicating that the presence of IDG also inhibited the inward Ca2+ influx.

As you pointed out, our experiments did not directly show that IDG's vasodilation is related to ROCC, but obtained such a result through indirect proof. The purpose of our entire experiment is to obtain the components that exert vasodilation effect in the ethyl acetate part of O. citriodorum and to study its vasodilation effect. This effect is more biased in vitro. And how IDG acts on ROCC, we will conduct in-depth research later. In the near future, we will give you a satisfactory answer.

Point 12. Figure 7 shows that IDG acts in VDCC, ROCC, KCa channels and also on exit of Ca2+ from endoplasmic reticulum. In my opinion, these are only possibilities or hypothesis, because the results shown does not demonstrate that these all are targets of IDG.

Response: Thank you for pointing out our problem. As answered in the previous question, the focus of our experiment is to isolate the components with vasodilating effect from the ethyl acetate part of O. citriodorum and explore the ways in which it works. Through vascular ring experiments and cell experiments, we can get the conclusion that the vasodilation effect of IDG may related to VDCC, ROCC, and KCa, but the direct role of IDG on these channels is lacking. Based on your comments, we will show that "the vasodilation effect of IDG was related to the VDCC, ROCC, and KCa ", not "the effect through the VDCC, ROCC, and KCa."

Point 13.The information concerning A7r5 cells culture is mentioned in section 4.12 and not in 4.3 (media, passage, timing…). In my opinion this is not convenient.

Response: We have listed the information about cell culture in a separate section based on your comments.

Point 14. - In this sense, this section has 13 subsections. In my opinion, there are only 3 types of experiments (extraction, contractility and intracellular calcium), 2 sample types (aorta rings and A7r5 cells), 1 set of chemicals (drugs and reagents) and statistics. Takin this in account, my opinion is that 7-8 subsections will be sufficient. The authors have 6 subsections starting by “Vasorelaxant Effects of..” or “Effects of…” that could be unified in 2 or maximum 3 subsections.

Response: Based on your comments, we have referenced some similar articles in the journal and adjusted the sections.

Point 15. I agree with the vasodilation and “endothelium-independent” effect of IDG, and also with intracellular reduction in calcium (this was demonstrated by the data). But the authors conclude “…involves two mechanisms: (1) inhibition of the entry of extracellular Ca2+ by blocking ROCC and VDCC and activating KCa, and (2) reducing [Ca2+]in release via inhibiting ROCC.” I think this are hypothesis to be confirmed because they are not completely demonstrated by the data.

Response: We agree with you. We lack enough experiments and data to prove how the vasodilation effect of IDG was achieved through ROCC or VDCC, we can only reflect that there is a relationship between this relaxation and the ROCC or VDCC pathways in our experiments. We have modified this section based on your comments.

Point 16. - Some conclusions concerning the presence of IDG in O. citriodorum and the scientific demonstration about the potential use of this natural product are missing. Also the potential of the plant to extract this and other “new drugs”.

Response: We have supplemented and modified the conclusions based on your comments